# Sage (*Salvia officinalis* L.) Essential Oil as a Potential Replacement for Sodium Nitrite in Dry Fermented Sausages

**Branislav Šojić [1], Vladimir Tomović [1,*], Jovo Savanović [1,2], Sunčica Kocić-Tanackov [1], Branimir Pavlić [1], Marija Jokanović [1], Ardea Milidrag [3], Aleksandra Martinović [4], Dragan Vujadinović [5] and Milan Vukić [5]**

[1] Faculty of Technology Novi Sad, University of Novi Sad, Bulevar cara Lazara 1, 21000 Novi Sad, Serbia; sojic@tf.uns.ac.rs (B.Š.); jovosavanovicdimdim@gmail.com (J.S.); suncicat@uns.ac.rs (S.K.-T.); bpavlic@uns.ac.rs (B.P.); marijaj@tf.uns.ac.rs (M.J.)

[2] "DIM-DIM" M.I. d.o.o, Svetosavska bb, 78252 Trn-Laktaši, Bosnia and Herzegovina

[3] Faculty of Medicine, University of Kragujevac, Svetozara Markovića 69, 34000 Kragujevac, Serbia; ardea304@gmail.com

[4] Faculty for Food Technology, Food Safety and Ecology, University of Donja Gorica, Donja Gorica, Oktoih 1, 81000 Podgorica, Montenegro; aleksandra.martinovic@udg.edu.me

[5] Faculty of Technology Zvornik, University of East Sarajevo, Karakaj 34a, 75400 Zvornik, Bosnia and Herzegovina; dragan.vujadinovic@tfzv.ues.rs.ba (D.V.); milan.vukic@tfzv.ues.rs.ba (M.V.)

* Correspondence: tomovic@uns.ac.rs; Tel.: +381-214853704

**Abstract:** This study investigates the effects of sodium nitrite replacement by the sage essential oil (SEO), on the physico-chemical, microbiological and sensory quality of dry fermented sausages (DFS) during 225 days of storage. The SEO (0.00, 0.05 and 0.10 µL/g) was added in DFS batters formulated with different levels of pork back fat (15% and 25%) and sodium nitrite (0, 75 and 150 mg/kg). The inclusion of SEO had no negative impact on pH, color (instrumental and sensory) and texture parameters. Total plate counts were lower than 6 log CFU (colony forming units)/g in all samples throughout the storage. Furthermore, the addition of SEO at concentration of 0.05 µL/g provided acceptable TBARS (2-Thiobarbituric acid reactive substances) values (<0.3 mg MDA (malondialdehyde)/kg) in the samples produced with reduced levels of sodium nitrite (0 and 75 mg/kg) without negative alternations on sensory attributes of odor and flavor. Generally, our findings confirmed that the usage of SEO could be a good solution to produce healthier DFS with reduced levels of sodium nitrite.

**Keywords:** sage essential oil; sodium nitrite; fat; dry fermented sausages

## 1. Introduction

Dry fermented sausages (DFS) represent a high-value dry cured meat product in European countries considering their unique eating quality and significant health benefit [1]. DFS are manufactured by mixing minced meat and fat with table salt, spices and their extracts, sugars, additives, starter cultures, vitamins, fibers, etc. [2]. The manufacture of DFS includes the following phases: mincing of the fresh or frozen meat and fat, seasoning and curing, stuffing into casings (natural or collagen), fermenting, smoking and drying, followed by the ripening that provides the typical sensory properties for this type of meat products [2,3]. Due to absence of thermal treatment, relatively high level of fat (15–40%) and usage of diverse raw materials, DFS have a strong risk of microbial growth and lipid oxidation, which consequently reduce the shelf-life of the product and have a significant health hazard to customers [4].

In order to ensure the shelf-life stability, several food additives are used during the processing of DFS [5]. One of these are nitrites which are frequently applied as effective curing agents in DFS processing, mainly as antioxidant and antimicrobial agents [6–9]. Likewise, these additives enhanced the color and flavor of cured meat products [6]. However, it is well-known that interaction between nitrites and secondary amines in meat products leads

to the nitrosamines' formation [5,7], which have a significant oncogenic risk potential for humans [8]. Hence, one of the main challenges for the meat industry is to find a functional replacement for nitrites (sodium and potassium nitrite) in cured meat products [9]. Many studies suggested that essential oils, obtained from a wide spectrum of plants (medicinal and aromatic), could be used as natural additives and potential alternatives for synthetic antioxidants, including nitrites. Furthermore, essential oils from different plant materials have attracted lots of attention from customers and the food industry owing to the wide range of bioactive compounds which possess both a significant health benefit [10–12] and strong antioxidant and antimicrobial potential in meat processing [3,4,13–15]. In addition, our earlier studies exposed that the application of essential oils had a positive effect on the color (increasing redness) of meat products produced with different levels of sodium nitrites [4,15,16]. It should be highlighted that bioactive compounds, present in essential oils (e.g., terpenoids, polyphenols), in complex interaction with myoglobin enhanced the color of meat products [16]. These results could be attributed to a lower oxidation of the iron atom within the heme group, and in consequence, a reduced formation of methemyoglobin [17].

Sage (*Salvia officinalis* L.) is a medicinal plant belonging to the *Lamiaceae* family, with strong antioxidant and antimicrobial potential [14]. Monoterpene ketones (camphor, *α*-thujone and *β*-thujone) are the most abundant terpenoid compounds of sage essential oil (SEO), however, significant antioxidant and antimicrobial activity of SEO has been primarily linked to a presence of diterpene polyphenols (carnosol, epirosmanol and carnosic acid) [18]. Even though sage has been traditionally used for culinary purposes, it has been reported that this plant and its formulations (e.g., essential oil, extracts, decoctions, etc.) exhibit a wide spectrum of biological activities such as antioxidant, antibacterial, antiproliferative and anticarcinogenic properties [18,19]. Several researchers assessed the application of SEO as an emerging additive in meat processing [14,20,21]. Fasseas et al. [20] determined that SEO possess a strong antioxidant potential in bovine ground meat during cold storage. Also, in our previous study [14], we found that essential oil obtained from sage by-products prolonged the shelf-life of fresh sausages. Moreover, Cegiełka et al. [21] reported that this essential oil powerfully suppressed lipid oxidation and inhibited the growth of a wide spectrum of bacteria (e.g., coliforms and *Enterobacteriaceae*) in mechanically separated chicken meat stored at −18 °C for 9 months. Finally, in our previous study [22], we determined that SEO improved the quality and safety of cooked pork sausages.

Regarding a strong antioxidative and antimicrobial potential, as well as a positive effect on the color formation, we hypothesized that SEO might be used as a replacement (partial or total) for sodium nitrite in DFS processing. Hence, several physico-chemical, microbiological and sensory attributes of DFS were investigated.

## 2. Materials and Methods

### 2.1. Sage Essential Oil

The sage essential oil (SEO), obtained using the steam distillation method, was purchased from the Herba doo producer (Belgrade, Serbia). SEO was retained in dark glass flasks at 4 °C pending investigates.

Gas Chromatography–Mass Spectrometry (GC-MS) Profile of Terpenoid Compounds

GC-MS analysis of SEO was determined according to the method previously described by Pavlić et al. [18]. Analysis was done on a gas chromatography system (Agilent GC890N, Santa Clara, CA, USA) coupled to mass spectrometer (Agilent MS 5759, Santa Clara, CA, USA) equipped with a HP-5MS column (30 m length, 0.25 mm inner diameter and 0.25 µm film thickness). Helium was used as a gas carrier with the flow rate of 2 mL/min. SEO was diluted in methylene chloride (approximately 1 mg/mL) and injected in volume of 5 µL with 30:1 split ratio. Temperature program in the chamber was: injector temperature 250 °C, detector temperature 300 °C, initial 60 °C, with linear increase of 4 °C/min up to 150 °C. Identification of compounds was based on retention times, mass spectra from the National

Institute of Standards and Technology (NIST) 05 and Wiley 7n database and comparison with standard compounds. Identification was confirmed for the selected terpenoinds by standard compounds which were initially dissolved in methylene chloride at different concentrations (1–500 µg/mL). Results were expressed as relative percentage (%).

*2.2. Preparation of DFS*

The samples of DFS were manufactured using lean pork shoulder and pork back fat according to a procedure previously described by Tomović et al. [4]. Sodium nitrite (SN) (0, 75 and 150 mg/kg) was added in the DFS batters formulated with two levels of pork back fat (15% and 25%). Finally, in each of the obtained batters, SEO was added in four concentrations (0.00, 0.01, 0.05 and 0.10 µL/g). A total of 24 batches (2 × 3 × 4) were produced (Figure 1). All batches were stuffed in collagen casings (⌀ 37 mm) and then subjected to a process of smoking and drying for 21 days (t = 14–16 °C; Relative humidity = 95–80%). Finally, the obtained sausages were vacuum-packaged and stored until 225 days at 15 ± 1 °C. Samples were collected at diverse stages of storage (0, 75, 150, 225 days) involving 3 sausages, randomly selected from each batch. A total of 288 samples were analyzed.

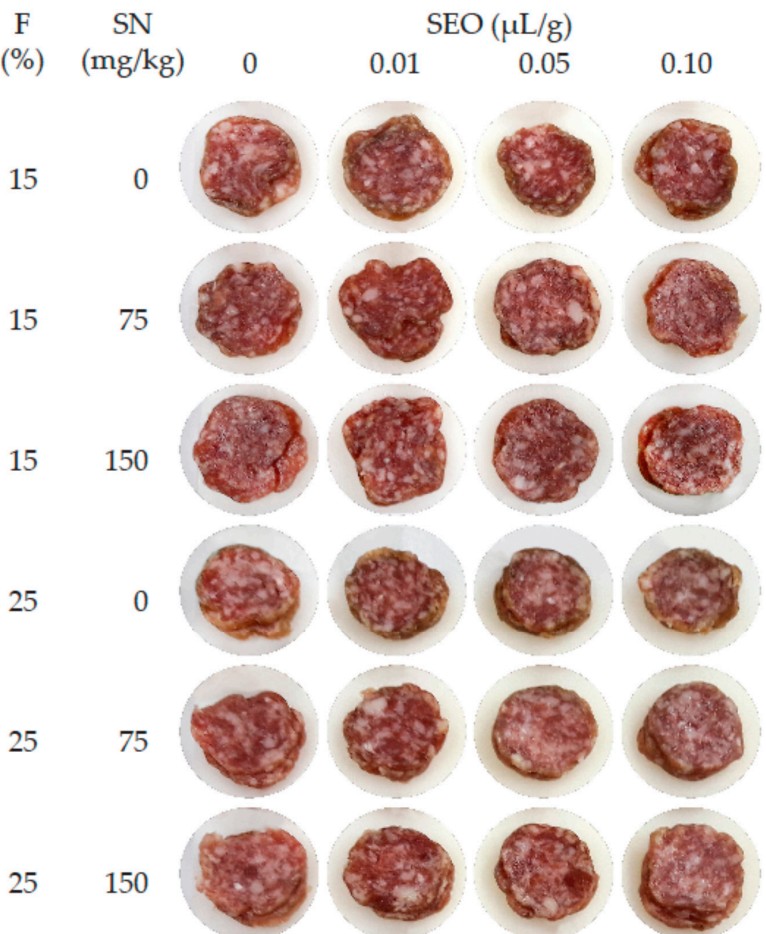

**Figure 1.** Photograph of the inside surfaces of the sausages at the end of storage. DFS—dry fermented sausages; F—fat; SN—sodium nitrite; SEO—sage essential oil; S—storage.

*2.3. Physico-Chemical Analysis*

The proximate chemical composition (moisture, protein, fat and ash) was determined according to standard ISO (International Organization for Standardization) procedure [23–26]. Residual nitrite (RN) content was assessed according to ISO 2918:1975 [27].

The pH was measured using a digital pH meter Testo 205 (Testo AG, Lenzkirch, Germany) equipped with a combined penetration tip. Residual nitrite (RN) content and pH value were determined for three samples from each group of DFS in duplicate.

The instrumental parameters of color (lightness: $L^*$, redness: $a^*$ and yellowness: $b^*$) were measured using a MINOLTA Chroma Meter CR-400 (Minolta Co., Ltd., Osaka, Japan) according to a previously described procedure in the study by Šojić et al. [16]. Color parameters were measured on cut surface of three samples from each group of DFS in triplicate.

The instrumental parameters of texture (Texture Profile Analysis—TPA): hardness (g), springiness, cohesiveness and chewiness (g), were determined according to a procedure previously described by Tomović et al. [4]. The degree of lipid oxidation was analyzed using the 2-Thiobarbituric acid reactive substances (TBARS) test, as described in our previous study [28]. The result was expressed as mg of malondialdehyde (MDA) per kg of sample. TPA and TBARS were determined for three samples belonging to each group of DFS in duplicate.

### 2.4. Microbiological Analysis

The subsequent microbiological analyses were performed [29–34]: total plate count—TPC (ISO 4833-1:2013), lactic acid bacteria—LAB (ISO 15214:1998), *Escherichia coli* (ISO 16649-2:2001), *Listeria monocytogenes* (ISO 11290-2:2017), *Salmonella* spp. (ISO 6579-1:2017) and sulfite-reducing clostridia count (ISO 15213:2003). Results were expressed as a log CFU (colony forming units)/g. All microbiological analyses were performed on three samples from each group of DFS in duplicate.

### 2.5. Sensory Analysis

Panelists (10 members) were skilled according to procedures described in ISO 8586: 2015 [35]. Sensory quality (color, odor and flavor) of DFS were examined in a Laboratory for sensory analysis of Faculty of Technology Novi Sad (ISO 8589:2007) [36] using the Difference-from-control test [37]. Firstly, panelists were asked to assess the control sample (without SEO and with the analogous levels of fat and sodium nitrite) and then to rate the difference between coded samples and the control one on a scale from 0 to 6, where 0 = no difference, 1 = very slight difference, 2 = slight/moderate difference, 3 = moderate difference, 4 = moderate/large difference, 5 = large difference and 6 = very large difference.

### 2.6. Statistical Analysis

The statistical program STATISTICA 13.0 (TIBCO Software Inc., Palo Alto, CA, USA) was used for data analyses. The experiment was performed in three independent replicates, included in the model as a random term. The differences between replicates were not significant ($p > 0.05$) for all physico-chemical, microbiological and sensory parameters. The main effects (fat content, nitrite content, SEO content and storage day) were compared. All data were expressed as mean value with their standard error (SE). The two-way, three-way and four-way interactions between these effects were also tested. Differences among means were compared according to t-test and Duncan's multiple range test ($p < 0.05$).

### 3. Results and Discussion

#### 3.1. Chemical Profile of SEO

Initially, GC-MS was performed for chemical characterization of SEO and terpenoids content with their respective retention times given in Table 1. Results suggested that *α*-thujone (35.50%) was the most dominant terpenoid in the SEO sample. It was in agreement with the literature [38]. Several other compounds detected in particularly high content were camphor (20.73%), eucalyptol (12.13%), *β*-thujone (6.57%), borneol (4.26%) and bornyl acetate (2.42%), suggesting that oxygenated monoterpenes were the major terpenoid subgroup present in SEO. Similar chemical profile of these compounds in SEO, as well as substantially high content of oxygenated monoterpenes, was observed by Radivojac et al. [39].

**Table 1.** Chemical profile of sage essential oil determined by gas chromatography–mass spectrometry.

| Compound | RT [1] (min) | Relative Percentage (%) |
|---|---|---|
| Sabinene | 4.362 | 0.06 |
| β-Pinene | 4.440 | 1.66 |
| β-Myrcene | 4.685 | 0.44 |
| n.i. [2] | 4.812 | 0.04 |
| *p*-Cymene | 5.488 | 1.79 |
| d,l-Limonene | 5.580 | 1.19 |
| Eucalyptol | 5.674 | 12.13 |
| n.i. | 5.802 | 0.06 |
| Linalool oxide | 6.713 | 0.09 |
| n.i. | 7.173 | 0.24 |
| α-Thujone | 7.781 | 35.50 |
| β-Thujone | 8.016 | 6.57 |
| n.i. | 8.143 | 0.14 |
| n.i. | 8.231 | 0.06 |
| n.i. | 8.653 | 0.12 |
| Camphor | 8.868 | 20.73 |
| n.i. | 8.946 | 0.14 |
| Borneol | 9.480 | 4.26 |
| 4-Terpineol | 9.775 | 0.55 |
| *p*-Cymene-8-ol | 10.067 | 0.24 |
| n.i. | 10.20 | 0.18 |
| n.i. | 10.364 | 0.12 |
| n.i. | 10.507 | 0.33 |
| *trans*-Carveol | 11.080 | 0.12 |
| Bornyl formetanate | 11.274 | 0.19 |
| Carvone | 11.809 | 0.06 |
| n.i. | 12.716 | 0.15 |
| Bornyl acetate | 13.122 | 2.42 |
| Sabinyl acetate | 13.345 | 0.35 |
| Carvacrol | 13.855 | 0.10 |
| α-Copaene | 15.901 | 0.09 |
| Caryophyllene | 17.243 | 0.75 |
| n.i. | 17.843 | 0.22 |
| α-Humulene | 18.288 | 1.69 |
| α-Amorphene | 18.989 | 0.12 |
| Ledene | 19.530 | 0.13 |
| Myristicin | 20.412 | 1.04 |
| n.i. | 21.206 | 0.06 |
| Elemicin | 21.455 | 0.27 |
| Caryophyllene oxide | 22.113 | 1.18 |
| Viridiflorol | 22.411 | 1.61 |
| n.i. | 22.605 | 0.51 |
| Humulene oxide | 22.887 | 1.63 |
| n.i. | 23.414 | 0.11 |
| n.i. | 23.542 | 0.07 |
| Apiol | 24.947 | 0.18 |
| n.i. | 26.048 | 0.06 |
| n.i. | 34.189 | 0.25 |
| Total | | 100 |

[1] RT = Retention time (min), [2] n.i. = Not identified.

On the other hand, monoterpene hydrocarbons such as *β*-pinene (1.66%), *p*-cymene (1.79%) and d,l-limonene (1.19%), together with sesquiterpenes such as *α*-humulene (1.69%), caryophyllene oxide (1.18%), viridiflorol (1.61%) and humulene oxide (1.63%), were present in lower content, while all other compounds were detected in content lower than 1%. Even though climate effects, plant properties and geographical origin could affect chemical profile of SEO, hydrodistillation is still the recognized method of choice to produce pure

volatile oil with high content of oxygenated terpenoids with strong antioxidative potential [18]. Even though polyphenols and diterpene polyphenols were often designated as the most powerful antioxidants from sage, SEO rich in oxygenated monoterpenes has generally strong antioxidant activity. This action could be due to its predominant compounds ($\alpha$-thujone, camphor, eucalyptol and $\beta$-thujone) separately [40,41]. However, antioxidant capacity of EOs does not necessarily depend on the bioactive potential of their main components. Several studies suggested that bioactivity could be modulated, often towards synergism and additive effect, by the presence of minor EO compounds [40]. On the other hand, oxygenated monoterpenes from SEO were often identified as the more potent antimicrobials compared to the sage polyphenols. According to Delamare et al. [42], camphor, thujone and eucalyptol expressed strong antimicrobial activity towards *Aeromonas hydrophila*, *Aeromonas sobria*, *B. megatherium*, *B. subtilis*, *B. cereus* and *Klebsiella oxytoca* strains.

### 3.2. Proximate Chemical Composition and RN Content of DFS

Proximate chemical composition and RN content of DFS (without SEO) at the end of drying (21st day) are shown in Table 2. The obtained values of moisture (31.5–35.0%) and protein (22.9–28.9%) contents were in accordance with Serbian Regulations [43] for this type of meat product. Moreover, incorporation of pork back fat at the levels of 15% and 25% affected final fat contents in intervals of 31.1–33.4% and 35.5–37.4%, respectively. The RN content declined quickly during the drying process (8.1–14.4 mg/kg), probably as the consequence of the interaction of meat pigments (e.g., myoglobin) with nitrites and other compounds [6].

**Table 2.** Proximate chemical composition and RN content of DFS.

| F (%) | SN (mg/kg) | Moisture (%) | Protein (%) | Fat (%) | Ash (%) | RN (mg/kg) |
|---|---|---|---|---|---|---|
| 15 | 0 | 32.4 ± 0.0 | 28.5 ± 0.0 | 31.1 ± 0.6 | 5.7 ± 0.0 | n.i. |
|  | 75 | 32.6 ± 0.1 | 28.9 ± 0.1 | 32.1 ± 0.2 | 5.6 ± 0.0 | 9.2 ± 0.1 |
|  | 150 | 31.5 ± 0.0 | 28.6 ± 0.1 | 33.4 ± 0.4 | 5.8 ± 0.0 | 14.4 ± 0.0 |
| 25 | 0 | 31.7 ± 0.1 | 22.9 ± 0.2 | 37.4 ± 0.1 | 5.8 ± 0.0 | n.i. |
|  | 75 | 35.0 ± 0.0 | 22.9 ± 0.3 | 35.5 ± 0.2 | 5.0 ± 0.0 | 8.1 ± 0.0 |
|  | 150 | 33.1 ± 0.0 | 23.6 ± 0.4 | 36.0 ± 0.2 | 5.2 ± 0.0 | 13.8 ± 0.0 |

DFS—dry fermented sausages; RN—residual nitrite; F—fat; SN—sodium nitrite; n.i.—not identified. Means ± Standard Error.

### 3.3. pH, Instrumental Parameters of Color and TBARS Values of DFS

pH values of DFS are shown in Table 3. The fat content had a significant ($p < 0.05$) effect on the pH values. It was in accordance with the study of Mora-Gallego et al. [44] and Fonseca et al. [45], who determined that increase of fat content decreases pH in fermented sausages. This phenomenon could be the result of the more intensive lipolysis in fermented sausages produced with higher fat contents [44]. The sodium nitrite addition (0, 75 and 150 mg/kg) did not significantly ($p > 0.05$) affect the pH values. Regarding SEO, the pH values of the samples produced with SEO addition were significantly ($p < 0.05$) higher compared to their counterpart. It could be the result of the chemical profile of SEO. Similarly, the results of our previous study [14] suggested that SEO increases pH in fresh pork sausages. Moreover, Zhang et al. [46] observed that bioactive compounds (e.g., polyphenolics) contribute to the growth of ammonia-producing bacteria and consequently increase the pH values of DFS. Moreover, pH values inconsistently increased ($p < 0.05$) during 225 days of storage. The increase of pH values during storage is in relation to the formation of amino acids, peptides and amines, and simultaneous reduction of lactic acid compounds during process of proteolysis in DFS [47,48]. The two-way (F × SEO, SN × SEO, S × SEO), three-way (F × SN × S, F × S × SEO) and four-way (F × SN × S × SEO) interactions significantly ($p < 0.05$–0.001) affected the pH values (Table 4). pH fluctuated in intervals from 5.18 (F = 25%, SN = 75 mg/kg, S = 0 days, SEO = 0.10 µL/g) to 5.66 (F = 15%, SN = 0 mg/kg, S = 75 days, SEO = 0.01 µL/g). The relatively high pH values could be the

consequence of relatively lower lactic acid bacteria (LAB) counts (<7 log CFU/g) (Table 5). Similar findings were reported by Nikolić et al. [2] and Ozaki et al. [7].

**Table 3.** pH, instrumental parameters of color and 2-Thiobarbituric acid reactive substances (TBARS) values of DFS.

| | pH | L* | a* | b* | TBARS (mg Malondialdehyde/kg) |
|---|---|---|---|---|---|
| | | | F (%) | | |
| 15 | 5.49 ± 0.01 [a] | 47.8 ± 0.2 [b] | 14.0 ± 0.1 [a] | 8.18 ± 0.08 [a] | 0.14 ± 0.01 [a] |
| 25 | 5.37 ± 0.01 [b] | 52.7 ± 0.2 [a] | 13.0 ± 0.1 [b] | 7.91 ± 0.07 [b] | 0.16 ± 0.01 [a] |
| *p* | <0.001 | <0.001 | <0.001 | <0.008 | 0.079 |
| | | | SN (mg/kg) | | |
| 0 | 5.42 ± 0.01 [a] | 50.7 ± 0.3 [a] | 13.5 ± 0.2 [a] | 7.78 ± 0.09 [c] | 0.19 ± 0.01 [a] |
| 75 | 5.43 ± 0.01 [a] | 50.3 ± 0.3 [a] | 13.5 ± 0.1 [a] | 8.04 ± 0.08 [b] | 0.13 ± 0.01 [b] |
| 150 | 5.44 ± 0.01 [a] | 49.7 ± 0.3 [a] | 13.4 ± 0.1 [a] | 8.32 ± 0.09 [a] | 0.12 ± 0.01 [b] |
| *p* | 0.666 | 0.054 | 0.897 | < 0.001 | < 0.001 |
| | | | SEO (μL/g) | | |
| 0 | 5.37 ± 0.01 [c] | 51.2 ± 0.4 [a] | 13.3 ± 0.2 [a] | 7.80 ± 0.09 [c] | 0.20 ± 0.01 [a] |
| 0.01 | 5.48 ± 0.01 [a] | 49.7 ± 0.3 [b] | 13.7 ± 0.2 [a] | 8.28 ± 0.10 [a] | 0.15 ± 0.01 [b] |
| 0.05 | 5.45 ± 0.01 [ab] | 49.8 ± 0.4 [b] | 13.7 ± 0.2 [a] | 8.16 ± 0.11 [ab] | 0.13 ± 0.01 [b] |
| 0.10 | 5.41 ± 0.01 [b] | 50.3 ± 0.3 [b] | 13.3 ± 0.2 [a] | 7.95 ± 0.09 [bc] | 0.12 ± 0.01 [b] |
| *p* | <0.001 | 0.005 | 0.044 | 0.003 | <0.001 |
| | | | S (day) | | |
| 0 | 5.32 ± 0.01 [c] | 51.0 ± 0.3 [a] | 13.0 ± 0.2 [c] | 7.57 ± 0.10 [c] | 0.03 ± 0.00 [d] |
| 75 | 5.50 ± 0.01 [a] | 49.9 ± 0.3 [b] | 13.5 ± 0.1 [b] | 8.00 ± 0.07 [b] | 0.11 ± 0.01 [c] |
| 150 | 5.43 ± 0.01 [b] | 50.3 ± 0.3 [ab] | 13.4 ± 0.2 [b] | 7.93 ± 0.09 [b] | 0.21 ± 0.00 [b] |
| 225 | 5.46 ± 0.01 [b] | 49.7 ± 0.4 [b] | 14.2 ± 0.2 [a] | 8.69 ± 0.11 [a] | 0.24 ± 0.01 [a] |
| *p* | <0.001 | 0.035 | <0.001 | <0.001 | <0.001 |

DFS—dry fermented sausages; F—fat; SN—sodium nitrite; SEO—sage essential oil; S—storage. Means ± SE with different letters [(a–d)] in the same column are significantly different ($p < 0.05$).

The instrumental parameters of color are depicted in Table 3. Fat content had a significant ($p < 0.05$) effect on the color of DFS. In the case of lightness and redness, the samples manufactured with lower fat content were darker and redder. It was in accordance with the findings of Lorenzo and Franco [49] and Fonseca et al. [45] for similar meat products. Also, Mora-Gallego et al. [44] reported that back fat particles in the DFS had a noticeable white color which conferred them a lighter appearance. The yellowness decreased while increasing the fat content, which agrees with the results of Rubio et al. [50] for similar meat products. The sodium nitrite addition significantly ($p < 0.05$) affected the yellowness of DFS. It was in accordance with the finding of Van Ba et al. [9] for fermented sausages. The inclusion of SEO significantly ($p < 0.05$) decreased the lightness and increased the yellowness. These phenomena could be the consequences of myoglobin interaction with the bioactive compounds of SEO, including terpenoids, phenolics, etc. [20]. Also, Zhang et al. [51] observed that the sage phenolic compounds could be oxidized into equivalent quinines, which resulted in formation of darker color of meat products. It was in strong agreement with results of our previous study [4,22]. The storage time significantly ($p < 0.05$) decreased in lightness and increased in redness and yellowness. Similar behavior of vacuum-packed DFS was reported by Pateiro et al. [52]. The increase of redness could be connected with the ability of *Staphylococcus* species (e.g., *S. carnosus* or *S. xylosus*) to produce the enzymes of metmyoglobin reductase and nitrate reductase, which possess a significant potential to modify brown metmyoglobin to a red myoglobin in cured meat products [53]. Moreover, the increase of yellowness could be related with the rancidity during long storage periods [54]. Also, the F × SN × S × SEO interaction had a significant ($p < 0.01$) effect on the lightness (Table 4). The lightness ranged in intervals from 44.2 (F = 15%, SN = 0 mg/kg, S = 225 days, SEO = 0.01 μL/g) to 56.6 (F = 25%, SN = 0 mg/kg, S = 0 days, SEO = 0 μL/g). The ob-

tained results were in accordance with literature data for similar meat products [2,7,9]. The two-way (N × S) and four-way (F × SN × S × SEO) interactions had a significant ($p < 0.01$–$0.001$) effect on the redness (Table 4). The lowest level of redness was determined in the sample: F = 25%, SN = 0 mg/kg, S = 0 days, SEO = 0 μL/g (9.38), while the highest level of redness were determined in the sample: F = 25%, SN = 0 mg/kg, S = 225 days, SEO = 0.05 μL/g (16.1). The obtained results suggested that SEO enhanced the redness of DFS produced without sodium nitrite. Accordingly, in our previous studies [15,16], we determined that essential oils obtained from coriander and organic peppermint improved the redness of cooked sausages manufactured with reduced levels (0 and 50 mg/kg) of sodium nitrite. The following interactions: F × SN, F × S, SN × S, SN × SEO, SN × S × SEO and F × SN × S × SEO, had a significant ($p < 0.05$–$0.001$) effect on the yellowness. The yellowness ranged in intervals from 5.89 (F = 25%, SN = 0 mg/kg, S = 0 days, SEO = 0.10 μL/g) to 10.02 (F = 15%, SN = 75 mg/kg, S = 225 days, SEO = 0.05 μL/g). The obtained results were in strong agreement with the literature data [2,9] for DFS.

　　TBARS values of DFS are presented in Table 3. Fat content did not significantly ($p > 0.05$) affect the TBARS values. The addition of sodium nitrite had a significant ($p < 0.05$) effect to decrease TBARS values. This was in accordance with strong antioxidative potential of this additive [6]. Also, SEO significantly ($p < 0.05$) decreased the TBARS values. It was in accordance with our previous study [14,22], where we found that SEO, in similar concentrations, had a strong antioxidative potential in processing of fresh [14] and cooked [22] sausages. Moreover, no significant differences ($p > 0.05$) were found among SEO-treated samples, hence the lowest concentration of SEO (0.01 μL/g) would be appropriate to enhance oxidative stability of DFS. Due to complex chemical profile with numerous bioactives diverse in volatility, polarity and bioactivity, several extraction procedures could be applied in order to provide selectivity towards target compounds. According to Radivojac et al. [39], the hydrodistillation procedure (conventional or microwave-assisted) could be an excellent tool to produce volatile SEO with high selectivity towards mono- and sesqui-terpenes. A recent study suggested that sage extracts rich in polyphenols and SEO with high content of terpenoids exhibit strong antioxidant activity in in vitro model systems and cell culture, as well as in in vivo studies on animals [55]. It has been suggested that SEO components seem to show a main role expression of antioxidant potential. Poulios et al. [55] suggested that *α*-thujone, *β*-thujone, camphor, linalool and eucalyptol are oxygenated monoterpenes mostly responsible for this action. Furthermore, presence of these compounds in mixture could provide synergistic effects, contributing to improved antioxidant action compared to effects caused by each compound separately [56]. As expected, TBARS values significantly ($p < 0.05$) increased throughout storage, probably as the consequence of lipid oxidation. The following interactions: F × S, SN × S, S × SEO, F × SN × S, SN × S × SEO and F × SN × S × SEO, significantly ($p < 0.05$–$0.001$) affected the TBARS values (Table 4). TBARS values ranged in intervals from 0.01 mg MDA/kg (F = 15%, SN = 0 mg/kg, S = 0 days, SEO = 0.05 μL/g) to 0.40 mg MDA/kg (F = 25%, SN = 0 mg/kg, S = 225 days, SEO = 0.05 μL/g). Melton [57] marked the TBARS values under 0.3 mg MDA/kg as a threshold for meat rancidity. In order to suppress the lipid oxidation, the synthetic (sodium nitrite) and natural additives (SEO) were used. The following combinations of sodium nitrite and SEO added in the DFS batters formulated with two levels of back fat enable TBARS values < 0.3 mg MDA/kg: F = 15–25%, SN = 150 mg/kg, S = 0–225 days, SEO = 0–0.10 μL/g; F = 15–25%, SN = 75 mg/kg, S = 0–225 days, SEO ≥ 0.05 μL/g and F = 15%, SN = 0 mg/kg, S = 0–225 days, SEO ≥ 0.05 μL/g. The obtained results suggested that the SEO (≥0.05 μL/g) in interaction with reduced level of sodium nitrite (75 mg/kg) decreased the lipid oxidation in DFS produced with both levels of back fat, 15% and 25%. Moreover, it should be highlighted that total replacement of sodium nitrite (0 mg/kg) by SEO (≥0.05 μL/g) provided satisfying oxidative stability (TBARS < 0.3 mg MDA/kg) in DFS produced with 15% of fat. SEO could express its antioxidant action through different mechanisms; however, prevention of lipid peroxidation is highlighted as the most important aspect for exhibiting protective effects in meat products.

**Table 4.** The effect of two-way, three-way and four-way interactions among processing parameters on the quality of DFS, expressed as *p*-value.

| Interactions | pH | L* | a* | b* | TBARS | TPC | LAB | Hardness | Springiness | Cohesiveness | Chewiness | Color | Odor | Flavor |
|---|---|---|---|---|---|---|---|---|---|---|---|---|---|---|
| F × SN | 0.232 | 0.326 | 0.352 | 0.044 | 0.428 | 0.387 | 0.999 | 0.197 | 0.870 | 0.007 | 0.843 | <0.001 | 0.581 | 0.679 |
| F × S | 0.405 | 0.415 | 0.788 | 0.047 | 0.009 | 0.509 | 0.344 | <0.001 | <0.001 | <0.001 | <0.001 | <0.001 | <0.001 | 0.153 |
| SN × S | 0.312 | 0.380 | <0.001 | <0.001 | <0.001 | 0.417 | 0.051 | 0.285 | 0.033 | 0.008 | 0.110 | <0.001 | 0.508 | 0.559 |
| F × SEO | 0.005 | 0.350 | 0.067 | 0.466 | 0.629 | 0.237 | 0.124 | 0.284 | 0.005 | <0.001 | 0.700 | <0.001 | 0.774 | <0.001 |
| SN × SEO | 0.005 | 0.162 | 0.088 | 0.018 | 0.747 | 0.315 | 0.089 | 0.499 | 0.220 | 0.318 | 0.438 | <0.001 | <0.001 | 0.003 |
| S × SEO | 0.007 | 0.829 | 0.829 | 0.572 | 0.004 | 0.278 | 0.264 | 0.816 | 0.099 | 0.056 | 0.152 | <0.001 | <0.001 | <0.001 |
| F × SN × S | 0.002 | 0.750 | 0.173 | 0.210 | 0.025 | 0.628 | 0.478 | <0.001 | <0.001 | <0.001 | <0.001 | <0.001 | 0.523 | 0.775 |
| F × SN × SEO | 0.682 | 0.220 | 0.069 | 0.900 | 0.986 | 0.453 | 0.322 | 0.781 | 0.418 | 0.515 | 0.633 | <0.001 | 0.707 | 0.127 |
| F × S × SEO | 0.028 | 0.109 | 0.068 | 0.377 | 0.061 | 0.389 | 0.214 | 0.003 | 0.299 | 0.032 | 0.012 | <0.001 | <0.001 | <0.001 |
| SN × S × SEO | 0.522 | 0.699 | 0.607 | 0.003 | <0.001 | 0.118 | 0.217 | 0.994 | 0.117 | 0.206 | 0.901 | <0.001 | 0.758 | <0.001 |
| F × SN × S × SEO | <0.001 | 0.003 | 0.009 | 0.001 | <0.001 | 0.256 | 0.321 | <0.001 | 0.002 | 0.001 | <0.001 | <0.001 | 0.094 | 0.208 |

DFS—dry fermented sausages; F—fat content; SN—nitrite content; S—storage day; SEO—sage essential oil.

**Table 5.** Microbiological profile of DFS.

| | TPC [log CFU (Colony Forming Units)/g] | LAB [log CFU (Colony Forming Units)/g] |
|---|---|---|
| **F (%)** | | |
| 15 | 5.58 ± 0.11 [a] | 5.55 ± 0.14 [a] |
| 25 | 5.27 ± 0.13 [a] | 5.39 ± 0.13 [a] |
| $p$ | 0.083 | 0.440 |
| **SN (mg/kg)** | | |
| 0 | 5.46 ± 0.14 [a] | 5.63 ± 0.12 [a] |
| 75 | 5.50 ± 0.15 [a] | 5.63 ± 0.16 [a] |
| 150 | 5.31 ± 0.17 [a] | 5.15 ± 0.20 [a] |
| $p$ | 0.673 | 0.073 |
| **SEO (μL/g)** | | |
| 0 | 5.51 ± 0.20 [a] | 5.62 ± 0.14 [a] |
| 0.01 | 5.45 ± 0.14 [a] | 5.51 ± 0.24 [a] |
| 0.05 | 5.36 ± 0.14 [a] | 5.27 ± 0.23 [a] |
| 0.10 | 5.38 ± 0.20 [a] | 5.49 ± 0.20 [a] |
| $p$ | 0.921 | 0.658 |
| **S (day)** | | |
| 0 | 5.28 ± 0.15 [b] | 6.50 ± 0.06 [a] |
| 75 | 4.56 ± 0.11 [c] | 5.50 ± 0.14 [b] |
| 150 | 5.97 ± 0.12 [a] | 5.35 ± 0.13 [b] |
| 225 | 5.89 ± 0.15 [a] | 4.53 ± 0.18 [c] |
| $p$ | <0.001 | <0.001 |

DFS—dry fermented sausages; TPC—total plate count; LAB—lactic acid bacteria; F—fat; SN—sodium nitrite; SEO—sage essential oil; S—storage. Means ± SE with different letters [(a–c)] in the same column are significantly different ($p < 0.05$).

### 3.4. Microbiological Profile of DFS

The microbiological profile of DFS is presented in Table 5.

The fat content and sodium nitrite addition did not have a significant ($p > 0.05$) effect on total plate counts (TPC) and lactic acid bacteria (LAB) in DFS. In the case of natural antioxidants, the SEO addition had a tendency to decrease TPC and LAB growth, but the values among the samples were not significantly ($p > 0.05$) different. The obtained results suggested that further optimization is essential. Also, our previous study [14] suggested that the usage of novel extraction techniques (e.g., supercritical fluid extraction—SFE) could be a good tool to improve the antimicrobial potential of SEO. The higher antimicrobial effect of extract obtained by SFE could be related to a synergistic effect of oxygenated terpenes and diterpene polyphenols which could be co-extracted by SFE [14]. Regarding storage time, a significant ($p < 0.05$) effect on the growth of TPC and LAB was observed. Population of LAB decreased during the storage, and this was in accordance with the findings reported by Tomović et al. [4] and Sucu and Turp [58] for similar meat products. TPC decreased during the first 75 days of storage (4.56 log CFU/g), then increased ($p < 0.05$) until 150 days of storage (5.97 log CFU/g), remaining constant until the end of storage (5.89 log CFU/g). It can be highlighted that throughout the storage, average TPC was lower than 6 log CFU/g. Ozaki et al. [7] suggested that TPC under 6 log CFU/g could be considered as a threshold for quality and shelf-life of meat products. Also, all two-way, three-way and four-way interactions had no significant ($p > 0.05$) effect on the TPC and LAB counts. Moreover, the following foodborne pathogenic bacteria: *E. coli*, *L. monocytogenes*, *Salmonella* spp. and sulfite-reducing clostridia, were not detected in any DFS sample throughout the storage. Therefore, all DFS samples were safe for consumption according to requirements of European Commission Regulation: No 2073/2005 [59].

### 3.5. Instrumental Texture Parameters of DFS

Instrumental texture parameters (hardness, springiness, cohesiveness and chewiness) are presented in Table 6.

**Table 6.** Instrumental texture parameters of DFS.

|  | Hardness (g) | Springiness | Cohesiveness | Chewiness (g) |
|---|---|---|---|---|
|  | F (%) | | | |
| 15 | 7674 ± 99 [a] | 0.493 ± 0.003 [b] | 0.508 ± 0.002 [b] | 1935 ± 31 [a] |
| 25 | 5519 ± 66 [b] | 0.513 ± 0.003 [a] | 0.527 ± 0.003 [a] | 1489 ± 20 [b] |
| *p* | <0.001 | <0.001 | <0.001 | <0.001 |
|  | SN (mg/kg) | | | |
| 0 | 6577 ± 123 [a] | 0.497 ± 0.004 [a] | 0.504 ± 0.003 [b] | 1648 ± 34 [a] |
| 75 | 6608 ± 147 [a] | 0.504 ± 0.004 [a] | 0.522 ± 0.003 [a] | 1729 ± 39 [a] |
| 150 | 6598 ± 131 [a] | 0.507 ± 0.004 [a] | 0.527 ± 0.003 [a] | 1757 ± 37 [a] |
| *p* | 0.986 | 0.152 | < 0.001 | 0.093 |
|  | SEO (µL/g) | | | |
| 0 | 6269 ± 161 [a] | 0.508 ± 0.005 [a] | 0.523 ± 0.004 [a] | 1654 ± 41 [a] |
| 0.01 | 6815 ± 163 [a] | 0.499 ± 0.005 [a] | 0.508 ± 0.003 [b] | 1727 ± 46 [a] |
| 0.05 | 6688 ± 133 [a] | 0.504 ± 0.004 [a] | 0.516 ± 0.003 [ab] | 1743 ± 40 [a] |
| 0.10 | 6604 ± 156 [a] | 0.502 ± 0.004 [a] | 0.522 ± 0.003 [a] | 1722 ± 42 [a] |
| *p* | 0.078 | 0.526 | 0.011 | 0.469 |
|  | S (day) | | | |
| 0 | 5007 ± 93 [d] | 0.450 ± 0.004 [d] | 0.544 ± 0.004 [a] | 1212 ± 20 [d] |
| 75 | 6614 ± 107 [c] | 0.508 ± 0.003 [c] | 0.528 ± 0.003 [b] | 1759 ± 24 [c] |
| 150 | 7735 ± 140 [a] | 0.522 ± 0.003 [ab] | 0.500 ± 0.003 [c] | 2007 ± 34 [a] |
| 225 | 7014 ± 151 [b] | 0.532 ± 0.003 [a] | 0.498 ± 0.003 [c] | 1866 ± 45 [b] |
| *p* | <0.001 | <0.001 | <0.001 | <0.001 |

DFS—dry fermented sausages; F—fat; SN—sodium nitrite; SEO—sage essential oil; S—storage. Means ± SE with different letters [(a–d)] in the same column are significantly different ($p < 0.05$).

Fat content significantly ($p < 0.05$) affected the texture parameters of DFS. It was in accordance with the studies of Tomović et al. [4], Fonseca et al. [45] and Mora-Gallego et al. [60] who observed that increase of fat content decreased the hardness and chewiness of DFS. The sodium nitrite and SEO addition significantly ($p < 0.05$) affected the cohesiveness. Similarly, positive correlations among sodium nitrite addition and cohesiveness were registered by Tomović et al. [4] for dry fermented sausages and Dong et al. [61] for cooked sausages. In the case of natural antioxidants, Pateiro et al. [52] found that relatively high concentrations ($\geq$200 mg/kg) of natural antioxidants can affect the texture parameters of DFS. It could be explained that phenolic compounds from natural antioxidants (grape seed extract, chestnut extract and green tea extract) in interaction with protein thiols result in modification of water-holding capacity and decrease the hardness of fermented sausages [52]. Finally, the storage time had a significant ($p < 0.05$) effect on the texture parameters of DFS. During storage, the hardness and chewiness values showed an increase ($p < 0.05$) until the 150th day of storage, followed by a decrease ($p < 0.05$) until day 225. The decrease of the hardness and chewiness values could be related to lipid oxidation [62]. The following interactions: F × S, F × SN × S and F × SN × S × SEO, had a significant ($p < 0.01$–0.001) effect on all texture parameters (Table 4).

### 3.6. Sensory Analysis of DFS

Sensory panel results are presented in Table 7.

Despite the contents of fat, sodium nitrite and SEO, as well as storage time, having significant ($p < 0.05$) effects on the sensory attribute of color, the differences between the samples were very slight (<1). Also, all two-way, three-way and four-way interactions had

a significant ($p < 0.001$) effect on this sensory attribute. The inclusion of sodium nitrite (150 mg/kg) significantly ($p < 0.05$) affected sensory attribute of odor. Moreover, the SEO addition had a significant ($p < 0.05$) effect on the odor and flavor. SEO at a concentration of 0.10 μL/g provided atypical flavor (>3) for DFS, which could be related to relatively high percentage of oxygenated monoterpenes in the chemical profile of SEO [14]. Therefore, the optimization of SEO in DFS processing is essential. Also, it should be noticed that the addition of $α$-thujone is limited (under 0.5 mg/kg), owing to its toxicity [63]. Regarding the SEO used in concentrations below 0.10 mL/kg, the content of thujones in the final meat product was undoubtedly under the maximal allowed content. The storage time also had a significant ($p < 0.05$) effect on odor and flavor, but the differences among the samples were very slight (<1) and between from very slight to slight/moderate (<2), respectively. The following interactions: SN × SEO, S × SEO and F × S × SEO, had a significant ($p < 0.01$–0.001) effect for both odor and flavor (Table 4).

**Table 7.** Sensory parameters of DFS.

| | Color | Odor | Flavor |
|---|---|---|---|
| **F (%)** | | | |
| 15 | 0.30 ± 0.02 [b] | 0.57 ± 0.03 [a] | 1.29 ± 0.05 [a] |
| 25 | 0.93 ± 0.04 [a] | 0.62 ± 0.03 [a] | 1.36 ± 0.05 [a] |
| *p* | <0.001 | 0.342 | 0.301 |
| **SN (mg/kg)** | | | |
| 0 | 0.77 ± 0.04 [a] | 0.51 ± 0.04 [b] | 1.35 ± 0.06 [a] |
| 75 | 0.57 ± 0.03 [b] | 0.61 ± 0.04 [ab] | 1.35 ± 0.06 [a] |
| 150 | 0.50 ± 0.04 [b] | 0.66 ± 0.04 [a] | 1.27 ± 0.06 [a] |
| *p* | <0.001 | <0.023 | 0.613 |
| **SEO (μL/g)** | | | |
| 0 | 0.00 ± 0.00 [c] | 0.00 ± 0.00 [d] | 0.00 ± 0.00 [d] |
| 0.01 | 0.87 ± 0.05 [a] | 0.15 ± 0.02 [c] | 0.37 ± 0.03 [c] |
| 0.05 | 0.88 ± 0.05 [a] | 0.66 ± 0.04 [b] | 1.60 ± 0.04 [b] |
| 0.10 | 0.70 ± 0.04 [b] | 1.63 ± 0.05 [a] | 3.32 ± 0.05 [a] |
| *p* | <0.001 | <0.001 | <0.001 |
| **S (day)** | | | |
| 0 | 0.59 ± 0.04 [b] | 0.58 ± 0.05 [b] | 1.29 ± 0.08 [b] |
| 75 | 0.25 ± 0.02 [c] | 0.50 ± 0.05 [b] | 1.52 ± 0.08 [a] |
| 150 | 0.80 ± 0.04 [a] | 0.80 ± 0.05 [a] | 1.41 ± 0.07 [ab] |
| 225 | 0.81 ± 0.06 [a] | 0.50 ± 0.04 [b] | 1.08 ± 0.06 [c] |
| *p* | <0.001 | <0.001 | <0.001 |

DFS—dry fermented sausages; F—fat; SN—sodium nitrite; SEO—sage essential oil; S—storage. Means ± SE with different letters [(a–d)] in the same column are significantly different ($p < 0.05$).

## 4. Conclusions

The most abundant compound of SEO is $α$-thujone (35.50%). The lower fat content provided higher values for pH, redness, yellowness, hardness and chewiness. Sodium nitrite and SEO possessed a strong antioxidative potential in DFS. Moreover, the partial or total replacement of sodium nitrite by SEO ($\geq$0.05 μL/g) ensures TBARS values below 0.3 mg MDA/kg in the following samples: F = 15–25%, SN = 75 mg/kg, S = 0–225 days, SEO = 0.05–0.10 μL/g, and F = 15%, SN = 0 mg/kg, S = 0–225 days, SEO = 0.05–0.10 μL/g. Also, it should be highlighted that all samples had satisfying microbiological shelf-life throughout the storage (225 days). The addition of SEO at $\leq$0.05 μL/g provided a typical flavor for DFS. Therefore, the obtained results suggested that the SEO (0.05 μL/g) could be used as an effective sodium nitrite replacement in DFS processing.

**Author Contributions:** Conceptualization, V.T.; methodology, B.Š. and V.T.; software, V.T.; formal analysis, B.Š., J.S., S.K.-T., B.P. and M.J.; investigation, B.Š. and V.T.; resources, J.S., A.M. (Ardea Milidrag), D.V. and M.V.; writing—original draft preparation, B.Š. and B.P.; writing—review and editing, V.T., A.M. (Aleksandra Martinović), D.V. and M.V.; supervision, A.M. (Ardea Milidrag) and D.V.; project administration, V.T. and A.M. (Ardea Milidrag). All authors have read and agreed to the published version of the manuscript.

**Funding:** This research was funded by the Ministry of Education, Science and Technological Development, Republic of Serbia, under Grant 451-03-68/2020-14/200134. Also, this research has been done in liaison with the activities defined by the grant for the establishment and implementation of the research-innovation-scientific program "Centre of Excellence (CoE) for digitalization of microbial food safety risk assessment and quality parameters for accurate food authenticity certification (FoodHub)", financed by the Ministry of Science of Montenegro, under the Grant No. 01-3660/2.

**Institutional Review Board Statement:** Not applicable.

**Informed Consent Statement:** Not applicable.

**Data Availability Statement:** Data is contained within the article.

**Conflicts of Interest:** The authors declare no conflict of interest.

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
