# Peer review of "Sage (Salvia officinalis L.) Essential Oil as a Potential Replacement for Sodium Nitrite in Dry Fermented Sausages"

_processes, doi:10.3390/pr9030424_

Round 1
Reviewer 1 Report
Dear Authors,
below is a review of your article.
Review Report
re: Manuscript ID: processes-1084990
A brief summary
The aim of the research was to evaluate the possibility of using sage essential oil as the partial or total substitute for sodium nitrite in processing of dry fermented sausage. The sausages were produced in industrial conditions and than subjected to rippening process. The dry sausages differed in terms of the level of fat raw material (two levels were used), the level of sodium nitrite (three levels) and the amount of sage essential oil (three levels). A total of 24 batches were produced. As the sodium nitrite substitute commercial formulation of sage essential oil was used. The analytical scope of the work was very wide, because the physical, chemical, microbiological and sensory quality attributes of sausages were investigated. The obtained results allowed for the formulation of conclusions. On the basis of the obtained results, it was confirmed that also in the production of dry fermented sausage sage essential oil could be a good alternative for sodium nitrite, because the quality of the final product was acceptable.
Broad comments
This article fits well with the aim and scope of the journal.
The subject of the article is in line with the still current trend of improving the health quality of meat products by use of natural ingredients rich in bioactive components. The trend to abandon the use of chemical additives and replace them with ingredients of natural origin with similar properties is not new. Likewise, the aim of this study was to replace sodium nitrate with an ingredient of natural origin, better perceived in terms of health. However, it is still very important that the thus modified food product is safe for the consumer and acceptable in terms of sensory.
This article confirms the knowledge already available regarding the properties and the possibility of using sage essential oil in food. Strong antioxidative potential of sage essentail oil was confirmed. The novelty is the fact that dry fermented sausage was selected as the test product. Dry fermented sausages are a group of sausages quite popular in Europe. It is very diverse, because sausages produced in different regions of Europe are not identical in terms of the raw materials used and their relative proportions, stages of the production process, etc.
Advantages of the article:
- The article was prepared with great care. The language is professional and understandable.
- The article may be of interest to readers as interest in the natural ingredients in food continues.
- Use of commercial sage essential oil expected to be reproducible. I order to identify the active substances in the essential oil the compounds of sage essential oil have been analysed by use of chromatography system coupled to mass spectrometer.
- The sausages were produced under industrial conditions, which allowed to properly reflect the maturation process and evaluate the product.
- The statistical analysis of the results is very detailed, also including the interactions of the variables used.
- In addition to identifying the differences between the variants of sausages (including differences in color, texture parameters, chemical composition, the course of the lipid oxidation process), an attempt was made to explain the reasons for the differences and / or describe the mechanism of changes. For this purpose, available literature sources as well as the results of own research were used.
- As an element of the novelty, the assessment of the possibility of using sage essential oil as a substitute for sodium nitrite in the dry fermented sausage production process can be considered. It is also a qualitative and quantitative evaluation of the components of sage essential oil.
In my opinion, the article requires only a few minor corrections, mainly of an editorial nature.
Minor comments:
- Line 58 – „…(Salvia officinalis L.)…” - Please write the Latin name of the plant species in italics.
- Lines 186-187 – „The obtained values of moisture (31.5-34.9%) and protein (22.8-28.9%) contents…” - In Table 2, no value of 34.9% for the moisture content or a value of 22.8% for the protein content of the sausage was found. Please correct the values.
- Line 190 – „The RN content declined quickly during the drying process (8.13-14.4 mg/kg),…” - In Table 2, no value of 8.13 mg/kg for the residua nitrite was found. Please correct the value.
- Line 347 Table 6. – „N (mg/kg)” - Please center.
Best regards,
Reviewer
Author Response
- Line 58 – „…(Salvia officinalis L.)…” - Please write the Latin name of the plant species in italics.
- According to reviewer`s suggestion we wrote the Latin name of the plant species in italics.
- Lines 186-187 – „The obtained values of moisture (31.5-34.9%) and protein (22.8-28.9%) contents…” - In Table 2, no value of 34.9% for the moisture content or a value of 22.8% for the protein content of the sausage was found. Please correct the values.
- We corrected these values.
- Line 190 – „The RN content declined quickly during the drying process (8.13-14.4 mg/kg),…” - In Table 2, no value of 8.13 mg/kg for the residua nitrite was found. Please correct the value.
- We corrected this value.
- Line 347 – Table 6. – „N (mg/kg)” - Please center.
- We corrected this.
Yours sincerely,
Vladimir Tomović and authors
Reviewer 2 Report
I read the paper "Sage (Salvia officinalis L.) essential oil as potential replacement for sodium nitrite in dry fermented sausages" findining it very interesting.
Manu projects are detecting to use natural essential oils rather then artificial or chemical additives, so the study couldn't stand out for originality.
Anyway the study is well created and performed. I really appreciated the supplementary Table, I think it should be included in the paper.
Few suggestions are reported in the text attached.

Author Response
Reviewer #2:
- Anyway the study is well created and performed. I really appreciated the supplementary Table, I think it should be included in the paper.
- According to reviewer's suggestion, we moved Table S1 from supplementary material to the main text (revised manuscript – Table 4).
- Read also: Rizzo & Muratore International Journal of Clinical Nutrition & Dietetics 2020, 6: 149.
- We added one more reference in section Introduction.
- I suggest to move this paragraph in Results par. around 267 lines it should fit better.
- We moved this paragraph according reviewer's suggestion (revised manuscript – lines 279-284).
- I suggest to improve explanation, reporting the meaning for F (%) and N (mg/kg).
We corrected this.
Yours sincerely,
Vladimir Tomović and authors
Reviewer 3 Report
This is a poor quality product development paper which has little scientific merit in terms of the findings and novelty of research. This is mostly repetitive work, as plant based extracts are well known to be replacers for sodium nitrite in cooking applications. Introduction is poor and a reasoniong of why the study is performed, or what is the novelty is absent. Methodology is also poorly presented. Methods are described without even mentioning how the substrate was prepared for the particular methods like microbio analysis, physico-chemical analysis etc. Experimental design is also poorly presented, with focus on over-extending the work done by mention of "288 samples" which actually includes all time points, replicates and everything. However, the greatest drawback of the study is in terms of Results & Discussions. The process factors studied did not had significant effect on any of the variables, which is not suprising, and at the same time makes the research paper very unattractive. The paper does not deserve publication in Processes.
Author Response
Reviewer #3:
- This is a poor quality product development paper which has little scientific merit in terms of the findings and novelty of research. This is mostly repetitive work, as plant based extracts are well known to be replacers for sodium nitrite in cooking applications. Introduction is poor and a reasoniong of why the study is performed, or what is the novelty is absent. Methodology is also poorly presented. Methods are described without even mentioning how the substrate was prepared for the particular methods like microbio analysis, physico-chemical analysis etc. Experimental design is also poorly presented, with focus on over-extending the work done by mention of "288 samples" which actually includes all time points, replicates and everything. However, the greatest drawback of the study is in terms of Results & Discussions. The process factors studied did not had significant effect on any of the variables, which is not suprising, and at the same time makes the research paper very unattractive. The paper does not deserve publication in Processes.
- Thank you for your revision. The using of natural antioxidants, included essential oils are one of the main challenges in novel meat processing. Regarding strong preservative potential, we used sage essential oil as emerging additive and potential substitute for sodium nitrite in processing of dry fermented sausages. The sausage samples were manufactured according procedures described by Tomović et al. (2020). In our study, we displayed chemical profile of sage essential oil and physico-chemical characteristics, microbiological profile and sensory characteristics of dry-fermented sausages. GC-MS analysis of SEO was determined according to the method previously described by Pavlić et al. (2018). Basic chemical composition and microbiological profile were determined using standard ISO procedures. TBARS value, texture profile and sensory analyses were determined according procedures described in section Material and methods. The experiment was performed in three independent replicates, included in the model as a random term. The main effects (fat content, nitrite content, SEO content and storage day) were compared. The two-way, three-way and four-way interactions between these effects were also tested. Differences among means were compared according to t-test and Duncan's multiple range test (p<0.05).
According suggestion Reviewer's 1, Reviewer's 2 and Reviewer's 4, we decided to revised the manuscript.
Yours sincerely,
Vladimir Tomović and authors
Reviewer 4 Report
Dear author,
I read your manuscript, and my comments showed attached file.
I think the research about essential oil is important in area of meat processing.
Readers will find it easier to understand a more focused study design.
Sincerely yours,

Author Response
Reviewer #4:
- You describe it as "Page 2, Line 80; Regarding a strong antioxidative and antimicrobial potential, as well as positive effect on the color formation,...", which means that SEO supplies NO like sodium nitrite, and myoglobin in DFS for nitrosyl-myoglobin. Is it? If so, please refer the appropriate literature to explain the mechanism. Or are you focusing on the function of nitrite as an antioxidant or antibacterial agent, not as a color former? If so, I think the above expression is inappropriate.
- According to reviewer's suggestion, we described the effect of essential oil on the color of meat products (lines 57-63 – revised manuscript).
„It should be highlighted that bioactive compounds, present in essential oils (e.g. terpenoids, polyphenols), in complex interaction with myoglobin enhanced the color of meat products (Šojić et al., 2019). These results could be attributed to a lower oxidation of the iron atom within the heme group, and in consequence a reduced formation of MetMb” (Faustman et al., 2010).
Reference:
Šojić, B.; Pavlić, B.; Ikonić, P.; Tomović, V.; Ikonić, B.; Zeković, Z.; Kocić-Tanackov, S.; Jokanović, M.; Škaljac, S.; Ivić, M. Coriander essential oil as natural food additive improves quality and safety of cooked pork sausages with different nitrite levels. Meat Sci. 2019, 157, 107879.
Faustman, C.; Sun, Q.; Mancini, R.; Suman, S.P. Myoglobin and lipid oxidation interactions: Mechanistic bases and control. Meat Sci. 2010, 86, 86–94.
- I don't understand this study design very well. Nowhere is it explained why the two fat percentages in this study design are 15% and 25%.
- Experiments in this paper were performed in industrial conditions and according to applicable regulations and standards. According to the regulations of the Republic of Serbia, after drying, moisture and protein content in dry fermented sausages should be less than 35% and more than 22% respectively, which in the final product corresponds to about 35% of fat. Therefore, according to these requirements, one group of dry fermented sausages was formulated with 25% of fat added to the sausage mixture, i.e. with the highest possible fat content in the final product (about 35%). Another group of sausages was formulated with an idea to produce final product with lower fat content (about 30%), which corresponds to a lower moisture content and higher protein content, so 15% of fat was added to the sausage mixture.
- Is the sample of DFS ingredient pork, beef, or a mix of pork and beef? The place of purchase of the raw meat is listed, but the type of raw meat is not listed.
- We used lean pork meat and pork back fat (line 105 – revised manuscript).
- In table 1, you described the α-Thujoneas major compounds among SEO. Are there any of these ingredients (Table 1) that show antioxidant or antibacterial properties? If so, it should be mentioned in the results.
We corrected this section according reviewer's suggestion (lines 184-195 – revised manuscript).
- You make DFS without a microbial starter. Then, in the microbiological test, Staphylococcus aureus is not tested, why not analyze it?
- We agree with reviewer's suggestion that Staphylococcus aureus is one of the main foodborne pathogens., but according Regulation (EC) No 2073/2005 determination of Staphylococcus aureus in DFS is not mandatory.
- I have a question about the total plate counts. In your results, TPC is 4.56-5.97 log CFU/g. However, in my opinion, TPC is generally about 7-8 log CFU / g, but isn't your TPC a little small?
The relatively small total plate count could be a consequence of absence of starter culture and good hygiene procedures in industrial plant. In our opinion, using vacuum packaging had also a significant effect on reducing TPC during storage.
- In Table 4, N is written as sodium nitrite, but isn't it residual nitrite ion? Materials and methods describes how to measure RN (residual nitrite), but in Table 4, it is listed as sodium nitrite. I'm confused as to whether N is RN or sodium nitrite concentration.
- In revised manuscript we marked sodium nitrite as SN and residual nitrite as RN.
- Your pH is around 5.3-5.5, but I don't think the DFS pH is low enough to provide safety and storage. What do you think of this?
- In our opinion good hygiene procedures, moisture content (<35%) and vacuum packaging provided the safety of DFS. According Serbian legislation, minimum pH value for high quality dry fermented sausages is 5.3.
- I have some question about Statistical analysis (page 4). You use Duncan's newt in multiple comparisons, but I think turkey's multiples are more common these days. Moreover, you are doing three way and four-way as well as two-way, does this make sense? I found it difficult to understand these even when I read the manuscript. Isn't two-way enough?
- We agree with Reviewer's 4 that a two-way statistical analysis is sufficient. However, due to the comments of other reviewers (’’Reviewer's 1: The statistical analysis of the results is very detailed, also including the interactions of the variables used’’ and ’’Reviewer's 2: I really appreciated the supplementary Table, I think it should be included in the paper’’) it seems that they are very satisfied with the applied and presented statistical analysis (Table 4) so, that is why the statistical analysis was kept in the same format. If Reviewer's 4 insists we will delete the three-way and four-way statistical analysis from Table 4 and the main text.
Yours sincerely,
Vladimir Tomović and authors
Round 2
Reviewer 4 Report
Dear author,
I read your revised version manuscript, and confirm your modified part in your manuscript. I think your manuscript is easier to understand. In statistical processing, Duncan's multiple range test remained the same. I think Turkey karmer test are more popular than Duncan‘s in multiple comparison tests these days. I thought your manuscript could be improved better by using the Turkey karmer test, but you can leave it as Duncan.
Sincerely yours,
Author Response
Dear reviewer, thank you for all suggestions and comments.
Authors